# A Novel Mutation of *MSH2* Gene in a Patient with Lynch Syndrome Presenting with Thirteen Metachronous Malignancies

**DOI:** 10.3390/jcm12175502

**Published:** 2023-08-24

**Authors:** Ugne Silinskaite, Edita Gavelienė, Rokas Stulpinas, Ramunas Janavicius, Tomas Poskus

**Affiliations:** 1Faculty of Medicine, Vilnius University, LT-03101 Vilnius, Lithuania; 2Institute of Clinical Medicine, Faculty of Medicine, Vilnius University, LT-03101 Vilnius, Lithuania; 3Department of Pathology, Forensic Medicine and Pharmacology, Institute of Biomedical Sciences, Faculty of Medicine, Vilnius University, LT-03101 Vilnius, Lithuania; 4National Center of Pathology, Vilnius University Hospital Santaros Clinics, LT-08406 Vilnius, Lithuania; 5Department of Oncogenetics, Hematology, Oncology and Transfusion Medicine Center, Vilnius University Hospital Santaros Clinics, LT-08661 Vilnius, Lithuania; 6Department of Experimental, Preventive and Clinical Medicine, State Research Institute, Center for Innovative Medicine, LT-08406 Vilnius, Lithuania

**Keywords:** lynch syndrome, hereditary nonpolyposis colorectal cancer, metachronous cancer, multiple malignancies, tumor, surveillance, detection

## Abstract

Lynch syndrome (LS), also known as hereditary nonpolyposis colorectal cancer (HNPCC), accounts for 2–3% of all colorectal cancers. This autosomal dominant disorder is associated with a predisposition to endometrial, stomach, small bowel, pancreatic, biliary tract, ovary, urinary tract, brain, and skin tumors. Lynch syndrome is caused by the mutation of the *MLH1*, *MSH2* (*EPCAM*), *MSH6*, and *PMS2* genes. In this article, a case study of a 70-year-old female patient with Lynch syndrome is presented. Over a span of 30 years, the patient underwent multiple surgical procedures for a total of thirteen different malignancies. She was found to have a deleterious pathogenic gene *MSH2* (NM_000251.2) variant (mutation) c.1774_1775insT in the 12th exon. This variant, c.1774_1775insT, represents a novel finding, as it has not been previously reported in existing databases or literature. No other case of 13 metachronous tumors in a patient with Lynch syndrome was found in the literature.

## 1. Introduction

Lynch syndrome (LS), which also used to be called hereditary nonpolyposis colorectal cancer (HNPCC), is the cause of 2–3% of all colorectal cancers. It is an autosomal dominant disorder that can lead to many different extracolonic manifestations, including endometrial, stomach, small bowel, pancreatic, biliary tract, ovary, urinary tract, brain, and skin malignancies [1], as well as colorectal cancer (CRC). Inheritance of a single mutant allele from either parent is sufficient for the diagnosis of Lynch syndrome as it is an autosomal dominant disorder [2]. The primary genes associated with LS are *MLH1*, *MSH2*, *MSH6*, and *PMS2*. While *MLH3* and *PMS1* have also been implicated in the disorder, their precise roles remain less well defined [3]. All of these six genes, *MLH1*, *MSH2*, *MSH6*, *PMS2*, *MSH3*, and *MLH3*, play a role in mismatch repair (MMR) [4]. The mismatch repair (MMR) system is important for maintaining genome integrity. The MMR system can detect misincorporated nucleotides and fix these mistakes by an excision or resynthesis process [5]. Microsatellite instability can be corrected by the MMR system as well [5]. However, when a mutation in genes encoding MMR system proteins occurs, microsatellite instability can be detected, and the development of various cancers can be implicated. Microsatellite instability (MSI) is found in 10–15% of all colorectal cancer (CRC) cases [6]. A total of 3% of these cases are associated with Lynch syndrome [7].

Both germline and somatic mutations are found in MMR genes, where *MLH1* promoter hypermethylation is the most common somatic event (epimutation) [8,9]. In order to detect changes in MMR proteins, different techniques can be used such as immunohistochemistry (IHC), polymerase chain reaction (PCR), and large-panel next-generation sequencing [10]. MMR proteins function as dimers. *MLH1* forms a complex with *PMS2*, and *MSH2* forms a complex with *MSH6*. Consequently, mutations in *MLH1* result in the immunohistochemical loss of staining in both *MLH1* and *PMS2*. The same situation can be seen in mutations occurring in *MSH2* that will result in the loss of staining in *MSH2* and *MSH6*. However, mutations in *PMS2* and *MSH6* only affect the mentioned gene’s product [10].

HNPCC can be classified into Lynch syndrome I and II. LS I is mainly associated with susceptibility to early colorectal cancer while LS II can be associated with extracolonic malignancies in addition to colorectal cancer [11].

The vast majority, approximately 95%, of all known Lynch-syndrome-associated mutations are caused by mutations in *MLH1*, *MSH2*, and *MSH6* [12]. For individuals who are 70 years old, the cumulative risk of developing cancer in any organ varies. *MLH1* mutation carriers have a risk of about 71%, *MSH2* carriers 74.5–77%, and *MSH6* carriers 46.3–75% [13,14]. Furthermore, the risk of developing cancer associated with LS before the age of 70 is higher in *MLH1* and *MSH2* mutation carriers than in *MSH6* mutation carriers [14].

On average, the cumulative risks of colorectal cancer among *MLH1* and *MSH2* mutation carriers who are 70 years old are estimated to be approximately 34% and 47% for male carriers, respectively. Similarly, women with *MLH1* and *MSH2* mutations exhibit a similar cumulative risk of CRC, estimated at around 36% and 37%, respectively. The estimated risks for developing endometrial cancer in *MLH1* and *MSH2* mutation carriers are approximately 18% and 30%, respectively.

Lynch syndrome is often under-recognized [15]. Diagnosis of Lynch syndrome requires genetic testing, which can be prohibitively expensive. A lack of patients’ and doctors’ education about the disorder can be one more limitation to performing genetic testing. Consequently, not all patients showing potential symptoms of LS, such as colorectal cancer, undergo testing. Diagnostic criteria commonly used to identify individuals with a potential diagnosis of LS usually are Amsterdam II criteria, as they help to identify families with LS, and Bethesda guidelines which assist in selecting tumors for testing [16]. The Amsterdam II criteria are considered to be specific but not sensitive, whereas the revised Bethesda Guidelines are more sensitive but less specific. According to the American Society of Colon and Rectal Surgeons (ASCRS) guidelines, germline sequencing of the mismatch repair genes remains the gold standard for confirming the causative gene mutation for Lynch syndrome [17].

This article presents a case study of a seventy-year-old female patient with Lynch syndrome type II. The patient’s medical history dates to 1993 when she underwent her initial surgery for cancer, followed by ten subsequent surgeries for tumor removal.

## 2. Case

A case of a 70-year-old female patient with Lynch syndrome is presented in this article. Table 1 provides an overview of the patient’s surgical history. Figure 1 displays the pathology of selected colonic and extraintestinal tumors. An earlier report is presented elsewhere [18].

Cancer is a common cause of death in the patient‘s family. Two of her cousins from the mother’s side died from intestinal cancer at the age of 55 and 56. It is known that the patient’s mother passed away due to colorectal cancer at the elderly age exceeding 70 years. Another cousin from her mother’s side suffered from colorectal cancer, but after undergoing a successful operation at the age of 84, she continues to live until this day without any relapses. The patient’s grandmother from her father’s side passed away due to uterine cancer at the age of 80. It can be seen that the patient is not the first one in her family to suffer from intestinal, as well as uterine, cancer, and Lynch syndrome could be suspected for the relatives mentioned earlier. According to the patient, three more relatives died from cancer: a cousin from the mother’s side died due to prostate cancer when he was only 54 years old, and two uncles from her father’s side passed away because of lung cancer at the age of 61 and 75.

In May 1993, the patient underwent a jejunal resection due to a diagnosis of adenocarcinoma of the jejunum. Enlarged lymph nodes were observed in the middle section of the mesentery of the transverse colon. The surgical procedure involved the removal of 40 cm of the jejunum and the middle portion of the transverse colon. The pathological diagnosis of the case was a poorly differentiated adenocarcinoma (G3; pT3N0), yet no original report or slides were available for review at the time of preparing this manuscript.

In February 2001, eight years following the initial tumor removal, the patient was diagnosed with a second tumor located in the rectum, 10 cm from the anus. Rectal resection with nerve-preserving total mesorectal excision with colorectal anastomosis was performed. Pathological examination revealed a moderately differentiated adenocarcinoma with identifiable mucin production (G2, pT3N0).

In December 2003, the patient underwent subtotal gastrectomy (Billroth II) following a diagnosis of stomach cancer. The pathological report revealed a poorly differentiated adenocarcinoma with a single lymph node metastasis (G3; pT2N1).

In May 2008, a prophylactic colonoscopy was performed, during which a tumor of splenic flexure was identified. The patient underwent open splenic flexure resection. The pathology report confirmed the presence of a poorly differentiated adenocarcinoma (G3; pT3N0).

In May 2013, an abdominal and pelvic CT scan was performed for the patient, during which thickening of the small intestinal wall up to 1 cm was found. Consequently, an ileal resection was conducted to remove the tumor, along with 15 cm segments from both sides of the tumor. The pathological examination once again confirmed a diagnosis of a poorly differentiated adenocarcinoma, yet with partial (20%) mucinous differentiation and intravascular invasion (G3; pT4N0 L/V1).

In August 2014, the patient was diagnosed with ascending colon cancer, leading to a right hemicolectomy. Three synchronic tumors were revealed: a caecal G2 adenocarcinoma with partial (10%) mucinous differentiation (pT1N0), a G2 conventional adenocarcinoma of the ascending colon (pT2N0), and a G3 mucinous adenocarcinoma with a signet-ring cell component also located in the ascending colon (pT3N1a). Since only one lymph node had a metastasis of purely signet-ring cells, the G3 tumor was interpreted as N1a, while the other two tumors were considered N0 to avoid overstaging.

This time, a first mismatch repair protein (MMRP) immunohistochemical analysis was performed indicating the identical loss of *MSH2* and *MSH6* proteins in all three tumors. In addition, immunohistochemical analysis on the archived tissue of a 2003 poorly differentiated gastric adenocarcinoma (G3; pT2N1) was performed with a similar detected loss of *MSH2*/*MSH6*. The patient was referred for genetic counseling, and molecular genetic testing was performed using Illumina Miseq next-generation sequencing technology with targeted sequencing of colorectal-cancer-related genes (*MLH1*, *MSH2*, *MSH6*, *PMS2*, *EPCAM*, *APC*, *MUTYH*, *STK11*, *SMAD4*, *BMPR1A TP53*, *NTHL1*, *POLE*, *POLD1*, *MSH3*) included in the TruSight Cancer panel, complemented by bioinformatic analysis. A deleterious pathogenic gene *MSH2* (NM_000251.2) variant (mutation) c.1774_1775insT was detected. This alteration, located in coding exon 12 of the *MSH2* gene, results from an insertion of one nucleotide at position 1774, causing a translational frameshift with a predicted alternate stop codon (p.Met592Ilefs*6). This variation of the coding sequence is expected to result in the loss of function by premature protein truncation or nonsense-mediated mRNA decay and allele frequency not found in the normal population control GnomAD database. As such, this alteration is interpreted as a disease-causing mutation and classified as a pathogenic, Class 5 mutation according to ACMG criteria. This variant is novel, since it has not been previously reported in the ClinVar and InSight databases or the literature. Additionally, a heterozygous monoallelic *MUTYH* (NM_001128425.2) variant c.1187G>A (p.Gly396Asp) was detected, which is associated with *MUTYH*-associated polyposis only in bi-allelic (homozygous/compound heterozygous) [19]. These data confirmed Lynch syndrome diagnosis on a molecular level, clinically manifesting as type II. However, the therapeutic strategy did not change after the diagnosis, and resection of the affected organ continues to be the best suitable treatment method. In addition, the patient’s first presentation in the small bowel and relative sparing of the colon allowed for the long-term avoidance of short bowel syndrome. Subsequently, genetic counseling and predictive testing were recommended for the patient’s relatives, but as far as it is known, they never got tested.

The postoperative course was complicated by an anastomotic leak, necessitating two additional relaparotomies. The resulting enterocutaneous fistula and short bowel syndrome were treated with supplemental parenteral nutrition. Ultimately, the patient was left with 70 cm of the small bowel and sigmoid colon, along with the persisting enterocutaneous fistula.

During a colonoscopy in 2015, a polypectomy was performed. The pathological diagnosis indicated a sigmoid colon tubular adenoma with high-grade dysplasia.

In 2016, two more tubular adenomas in the duodenum, one of which contained high-grade dysplasia, were removed.

In 2017, an upper gastrointestinal (GI) endoscopy revealed a tubular adenoma with high-grade dysplasia in the small intestine, not amenable to endoscopic removal. Consequently, surgical exploration was undertaken, involving extensive adhesiolysis, resection of the ileocolic anastomosis, and closure of the enterocutaneous fistula. The patient experienced a successful recovery and did not require further parenteral nutrition.

In 2019, tubular adenomas of the small intestine were removed twice.

In June 2020, the patient underwent a biopsy with a subsequent left nephroureterectomy. The pathological examination once again revealed two synchronous tumors: an invasive high-grade urothelial carcinoma with focal myxoid stroma of the ureter (G3; pT2) and an invasive high-grade urothelial carcinoma with scattered signet-ring cells of the renal pelvis (G3; pT1). Both of these tumors exhibited a loss of *MSH2*/*MSH6* immunohistochemical expression, suggesting an association with Lynch syndrome. Kidney-sparing treatment was not possible, and a nephroureterectomy was performed.

Four months later, in October 2020, an enteroscopy was conducted, revealing the presence of duodenal carcinoma and a small intestine polyp. The polyp was successfully removed with biopsy scissors, but upon pathological examination, it was determined to be lymphangiectasia, rather than a tumor. In order to remove the carcinoma, local resection of the duodenum was performed with a final diagnosis of a moderately differentiated adenocarcinoma (G2; pT1b).

In 2020, sebaceoma of the scalp was detected and confirmed by the pathologists. Immunohistochemical analysis revealed the absence of *MSH2* and *MSH6* expression in the sebaceoma. The development of sebaceoma together with colorectal cancer and *MSH2* gene mutation correctly suggests Muir-Torre syndrome. However, cutaneous presentations in this patient are minimal in comparison to internal organ malignancies. Muir-Torre syndrome was not diagnosed because of overwhelming gastrointestinal presentation of tumors.

In 2021, endoscopic polypectomy was performed with the pathologist report showing Brunner gland polyps in the stomach as well as large intestine adenoma with high-grade dysplasia.

In 2022, a colon adenoma was detected and subsequently removed during a colonoscopy. Immunohistochemical analysis revealed the absence of *MSH6* expression.

In January 2023, a hysteroscopic polypectomy was carried out, revealing a well-differentiated endometrioid carcinoma harboring the same immunohistochemical *MSH6* loss. The patient underwent a total hysterectomy with a bilateral salpingectomy with a final tumor stage of pT1a.

Two months later, in 2023, a polyp in the sigmoid colon was detected and removed through a colonoscopic polypectomy.

In July 2023, a right ureteropyeloscopy was conducted on the patient due to high-grade urothelial carcinoma (G3, pT1N0M0).

## 3. Discussion

This study presents a case of a woman who over the period of 30 years underwent eleven operations for tumors associated with Lynch syndrome. As a result of multiple bowel resections and associated complications, she experienced malabsorption and developed short bowel syndrome. The patient required a stoma and relied on parenteral nutrition for a duration of approximately three years.

Colorectal cancer is often the initial malignancy observed in patients with Lynch syndrome. Colorectal cancer was the first tumor observed in 89% of affected males and 66% of affected females [20]. However, it was not the case for the patient as her first cancer at the age of 40 was located in the jejunum. Both the location and the young age are extremely rare [21], indicating a possible hereditary nature of the tumor. For *MLH1* or *MSH2* carriers, the risk of developing cancer in the small bowel is about 5% [22]. In patients with Lynch syndrome, small intestine cancers distribute by 33% in the jejunum, 43% in the duodenum, and 7% in the ileum [23].

Even though the study shows the 5-year survival rate for small intestine adenocarcinomas to be 35% in the United States population [21], the patient continues living until this day since her first adenocarcinoma in the small intestine (jejunum) in 1993. Interestingly, ASCRS [17] does not recommend routine screening of the small intestine.

The second tumor was adenocarcinoma in the rectum, diagnosed 8 years later. Colorectal cancer is usually the most common cancer found in Lynch syndrome carriers. European guidelines from the European Hereditary Tumour Group (EHTG) and European Society of Coloproctology (ESCP) [22], as well as ASCRS [11], advocate for extended surgery with ileosigmoidal/ileorectal anastomosis. It is preferable to standard resection in order to reduce the risk of metachronous CRC. European guidelines especially highlight it for *MLH1* or *MSH2* carriers. At the time of the surgery, the patient was not diagnosed with LS. Rectum resection with mesorectum excision was performed. Looking at the newest European guidelines [24], it was the best decision as MMR carriers with primary rectal cancer are recommended an anterior resection. Nonetheless, in 2008 (7 years later), carcinoma in the splenic flexure was found, and splenic flexure resection was performed. ASCRS newest guidelines for LS patients or people at risk recommend a colonoscopy every 1 to 2 years [17]. European guidelines recommend repeated colonoscopy for patients with *MLH1*, *MSH2*, and *MSH6* mutations every 2–3 years [24]. It is known that a colonoscopy every 3 years drastically decreases the risk of colorectal cancer and reduces overall mortality by 65% in HNPCC families [25]. The same European guidelines recommend for path *MLH1* or path *MSH2* carriers that surveillance colonoscopies be initiated at the age of 25 years. It was not known that the patient is a carrier of *MSH2* mutation, and therefore, she had her first colonoscopy many years later. Looking retrospectively, performing colonoscopies earlier for the patient may have avoided many surgeries and allowed the earlier diagnosis of LS.

At the age of 70, there is about a 5% risk for patients with LS to develop gastric cancer [20]. The patient‘s third tumor was in fact found in the stomach, but at the time, she was 50 years old. Some studies showed [26,27] that most gastric cancers associated with LS are of the intestinal type. One of those studies [26] revealed that *MSH2* carriers (9%) are at higher risk for gastric cancer than *MLH1* carriers (4.8%) and showed that *MSH6* carriers have not developed gastric cancer during their lifetime. Later in her life, the patient was identified as having mutations in *MSH2*. It is thought that gastric cancer in LS patients is more common in Asian countries, such as Japan and Korea, and Brazil than in Europe and North America [28]. ASCRS recommends screening patients at risk or with diagnosed LS using esophagogastroduodenoscopy with gastric biopsy of the antrum at age 30–35 years old [17].

The eighth tumor removal operation was a left nephroureterectomy because the tumors were found in the ureter and renal pelvis. LS patients have an increased risk for urothelial cancers in the upper urinary tract. *MSH2* mutation carriers are known to have a much higher risk of developing urothelial cancer than *MLH1* or *MSH6* carriers [29,30]. There appear to be no significant differences between sexes for developing urinary tract cancers in LS patients [30]. ASCRS recommends starting screening LS patients and people at risk with annual urinalysis at ages 30–35 years old [17].

In 2023, a total hysterectomy with bilateral salpingectomy was performed, and an endometrial malignant tumor was removed. Women diagnosed with LS have a 27—71% cumulative lifetime risk of developing endometrial cancer while in the general population, this risk is only 3% [1]. Endometrial cancer [20] was the first and only tumor at disease onset for 26% of the affected female *MLH1* and *MSH2* gene carriers. LS female patients carrying an *MSH6* mutation are at much more significant risk for endometrial cancer than those with *MLH1* and *MSH2* mutations [31]. The risk for endometrial cancer in female *MLH1* and *MSH2* mutation carriers is from 27% to 60%, while in *MSH6* mutation carriers, it increases from 60% up to 71% [1]. ASCRS [17] advocates for pelvic examination and endometrial sampling annually to diagnose endometrial cancer early and transvaginal ultrasound annually to diagnose ovarian cancer starting from 35 years of age.

The histological features suggestive of microsatellite instability in a colorectal adenocarcinoma are tumor-infiltrating lymphocytes (TILs), so-called Crohn’s-like peritumoral transmural lymphocytic reaction, extracellular mucin pools, medullary growth pattern or signet-ring cells (or otherwise poorly differentiated/undifferentiated areas), and a lack of tumor budding. The patient reported herein had a total of five colorectal tumors. The histology of the first one, a rectal carcinoma of 2001, was not reviewed since the slides were not available anymore. Yet, the description in the pathology report mentions both small areas of solid growth and mucin production, at least hinting at the possibility of an MMRP-deficient tumor. The second one, a splenic flexure carcinoma (Figure 1, panel 1a) was a nodular tumor composed of solid neoplastic cell sheets, or areas of discohesive cells with rhabdoid features and tumor-infiltrating lymphocytes—a classic MSI tumor pattern. The three simultaneous tumors of 2014 had mixed features: the G3 tumor with a single lymph node metastasis had both abundant extracellular mucin production and signet-ring cells (Figure 1, panel 1b), while the other two were more conventional gland-forming adenocarcinomas, yet extracellular mucins were also present in lesser quantities (Figure 1, panel 1c).

The histologic features that are specifically suggestive of MSI sebaceoma have not been extensively described, yet the very presence of a sebaceous neoplasm of the skin raises a possibility of Muir-Torre syndrome, a variant of Lynch syndrome. Yet this option was dismissed due to the overwhelming gastrointestinal presentation of tumors.

Urothelial MSI tumors share the presence of TILs with MSI colorectal carcinoma, but there are also distinct characteristics that can be suggestive of microsatellite instability. MSI urothelial carcinomas can have an inverted growth pattern, a higher grade, conspicuous nucleoli, and brisk mitotic activity; some of the tumors might have heterogeneous areas. The urothelial carcinoma described here was a G3 tumor with focal myxoid stroma, an admixture of signet-ring cells, and focal perineoplastic lymphocytic infiltration, thus exhibiting at least some features suggestive of MSI.

Lastly, the list of features suggestive of MSI endometrioid carcinoma is mostly composed of the ones described above and includes TILs, solid areas, mucinous differentiation, a higher grade, and tumor heterogeneity. Nevertheless, MSI endometrioid carcinomas often present as conventional G1 tumors as was in this case.

Short bowel syndrome (SBS) is a rare malabsorption condition because of the patient’s lack of small intestine [32]. Usually, symptoms of SBS can be noticed when a patient is left with less than 200 cm of the small intestine [33]. SBS goes hand in hand with a reduction in gut function and impaired absorption of micronutrients, water, and electrolytes. An intravenous supplementation is needed to get all the required nutrients. Abdominal operations account for up to 50 percent of all SBS cases. The patient has only 60 cm of small intestine left, underwent three large intestine operations, and suffers from short bowel syndrome. She used parenteral nutrition for 3 years (2014–2017). In 2017, a port-a-cath was removed, and now the patient is not dependent on parenteral nutrition. Even though short bowel can be expected in many LS patients, not a lot of literature exists on the topic.

There is no unified opinion on whether aspirin works as a chemopreventative drug and reduces the risk of colorectal cancer [34] or has no effect [1]. Both studies, however, agree that resistant starch was not beneficial for the chemoprevention of colorectal cancer. Another study states [35] that aspirin, ibuprofen, calcium, or multivitamins have a positive effect on decreasing colorectal cancer risk in MMR carriers, but large-scale, prospective, placebo-controlled prevention trials with these agents have failed to show reduced CRC incidence in non-Lynch syndrome populations. For females with LS oral contraceptives may be used as a chemopreventative drug [36,37] as it is believed it decreases the risk of endometrial and ovarian cancer. The patient was not advised or prescribed to use any kind of chemopreventative drug. Knowing this, it is recommended for female LS patients to use aspirin and oral contraception during reproductive age and afterward to undergo a prophylactic total hysterectomy and bilateral salpingo-oophorectomy (THBSO).

Even though data are still scarce, research [38] indicates that Lynch syndrome patients may benefit from immune-checkpoint-based therapies as no difference in the response rates has been reported between Lynch syndrome and sporadic MSI cancer patients. However, it is unknown why some LS patients do not respond to immune-checkpoint-based therapies. Studies on murines [39,40] showed that a tumor vaccine could prolong the overall survival of mice and improve the disease-free survival rate. In colon cancer associated with Lynch syndrome, a tumor vaccine can hinder the growth of tumor cells and increase the immunotherapy effect on this kind of cancer.

This study has inherent limitations due to its nature as a case report. However, this report presents a patient with a previously unknown type of mutation of the *MSH2* gene with multiple malignancies throughout the digestive and urogenital tracts and a possibility of a surgically induced short bowel syndrome. The tumors exhibit a relatively benign course, since none of the thirteen malignancies metastasized. No other report on a Lynch syndrome patient with thirteen malignant tumors was found in the PubMed database.

## 4. Conclusions

Female patients with *MLH1* and *MSH2* gene mutations face a 27–60% risk of developing endometrial cancer. Therefore, it is recommended that women at risk undergo a transvaginal ultrasound scan, endometrial biopsy, and pelvic examination starting at the ages of 30–35. LS patients should undergo colonoscopies every 1–2 years. While extensive colorectal surgery is considered the optimal treatment for CRC, there should be caution regarding extensive prophylactic gastrointestinal surgery due to the risk of developing short bowel syndrome. Aspirin has shown promising results as an effective preventative drug in many studies and could be recommended to the patient.

## Figures and Tables

**Figure 1 jcm-12-05502-f001:**
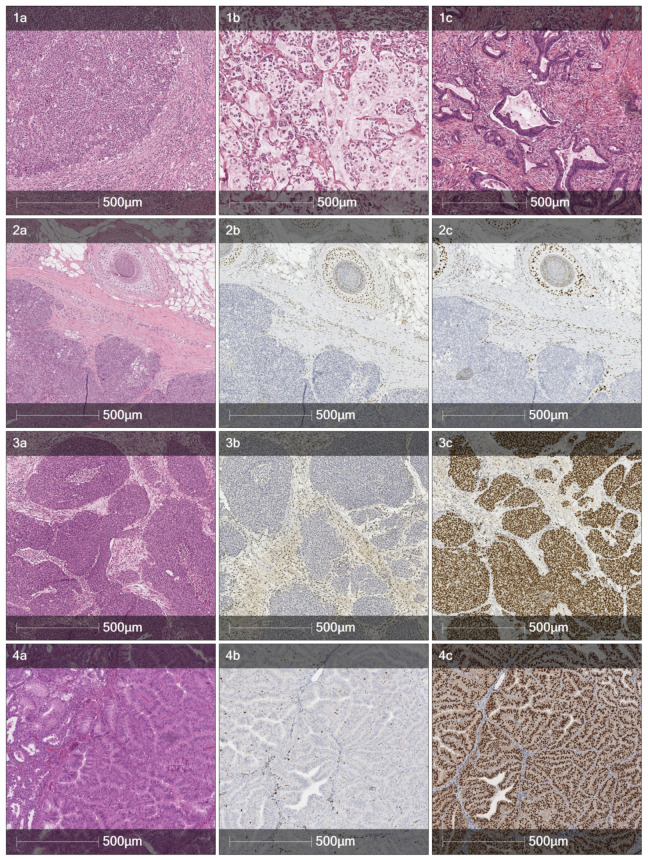
Pathology of selected colonic and extraintestinal tumors (10× magnification). First row: a diverse range of colon carcinomas observed over the years: (**1a**)—discohesive with rhabdoid features (2008), (**1b**)—mucinous with scattered signet-ring cells (2014), and (**1c**)—conventional low-grade (G2) adenocarcinoma (2014). Second row: sebaceoma of the scalp (2020): (**2a**)—tumor nests with basaloid cells and focal central sebaceous differentiation, (**2b**)—loss of nuclear *MSH2* expression in tumor cells, and (**2c**)—loss of nuclear *MSH6* expression in tumor cells. Third row: invasive high-grade urothelial carcinoma (2020): (**3a**)—irregular nests of the neoplastic urothelium, (**3b**)—loss of nuclear *MSH2* expression in tumor cells, and (**3c**)—strong positive nuclear GATA3 reaction supporting the diagnosis of urothelial carcinoma. Fourth row: well-differentiated (G1) endometrioid carcinoma (2022): (**4a**)—confluent glands with back-to-back crowding, (**4b**)—loss of nuclear *MSH2* expression in tumor cells, and (**4c**)—strong positivity for estrogen receptors characteristic of endometrioid carcinoma.

**Table 1 jcm-12-05502-t001:** Surgical history.

Year	Age at Diagnosis	Localization	Tumor Grade	Stage	Operation	MSI
1993	40	Jejunum	G3	pT3N0M0	Jejunal resection	No data
2001	48	Rectum	G2	pT3N0M0	Rectal resection with mesorectal excision	No data
2003	50	Stomach	G3	pT2N1M0	Subtotal gastric resection (Billroth II)	*MSH2/MSH6* (-)
2008	55	Colon	G3	pT3N0M0	Splenic flexure resection	No data
2013	60	Ileum	G3	pT4N0 L/V1	Ileal resection	No data
2014	61	Ascending colon	G2 G3	pT2N0M0 pT3N1aM0	Right hemicolectomy	*MSH2/MSH6* (-)
		Cecum	G2	pT1N0M0		*MSH2/MSH6* (-)
2017	64	Ileum	-	-	Ileal resection Reconstruction of ileocolic anastomosis	No data
2020	67	Left kidney	G3	pT1N0M0	Left nephroureterectomy	*MSH2/MSH6* (-)
		Ureter	G3	pT2N0M0		
		Duodenum	G2	pT1bN0M0	Duodenal resection	No data
2023	70	Uterus	G1	pT1a LVI0	Retroperitoneal total hysterectomy with bilateral salpingectomy	*MSH6* (-)
		Cervix				
		Ovaries				
		Fallopian tubes				
		Right kidney	G3	pT1N0M0	Right ureteropyeloscopy	No data

## Data Availability

The dataset used during the current study is available from the corresponding author on reasonable request.

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
