# Peer review of "A Novel Mutation of MSH2 Gene in a Patient with Lynch Syndrome Presenting with Thirteen Metachronous Malignancies"

_jcm, 2023, doi:10.3390/jcm12175502_

Round 1
Reviewer 1 Report
Poskus and colleagues present an interesting case in this manuscript; a 70 year old patient with 13 metachronous malignancies (starting from the age of 40), manifestations (colonic and extracolonic) of Lynch syndrome, associated with a novel pathogenic (?) variant in the 12th exon of MSH2 gene. The novel MSH2 variant and the very unusual clinical presentation are the main interesting points of this case report.
However, the report focuses too much on the general aspects of Lynch syndrome and less on the patient. Specific points:
1. Family history?
2. Therapeutic strategy? Did it change (and how?) after the results of the genetic testing?
3. Germline testing? Please describe the methodology and other findings of NGS testing. Which databases did you check to confirm that this finding is novel? What are the requirements that should be met to describe this finding as pathogenic with only one patient reported so far?
4. Have you reported this finding to databases like Clinvar?
5. The pathology images in the manuscript are very informative. Please discuss how the pathological characteristics of these tumors match (or not) the characteristics/phenotype that are suggestive of MMR-deficient tumors?
6. Please include MLH1/MSH6 immunohistochemistry staining.
7. Does the development of sebaceoma suggest Muir-Torre syndrome? Please discuss.
8. 2014 ascending colon resection with two cancers (one staged as N0, the other as N1). How is this possible given that the regional lymph nodes are the same for each tumor?
9. line 241 MLH6 mutation?
Minor editing and spell check is needed.
Author Response
- Family history?
(Please, see lines 121-132 in a file “jcm-2537243 - editing can be seen”) Cancer is a common cause of death in the patient‘s family. Two of her cousins from the mother’s side died from intestinal cancer at the age of 55 and 56. It is known that the patient’s mother passed away due to colorectal cancer at the elderly age exceeding 70 years. Another cousin from her mother’s side suffered from colorectal cancer but after undergoing a successful operation at the age of 84 she continues to live until this day without any relapses. The patient’s grandmother from her father’s side passed away due to uterine cancer at the age of 80. It can be seen that the patient is not the first one in her family to suffer from intestinal as well as uterine cancer and Lynch syndrome could be suspected for the relatives mentioned earlier. According to the patient, 3 more relatives died from cancer: a cousin from the mother’s side died due to prostate cancer when he was only 54 years old, and two uncles from her father’s side passed away because of lung cancer at the age of 61 and 75.
(Please, see line 191-193 in a file “jcm-2537243 - editing can be seen”) Subsequently, genetic counseling and predictive testing were recommended for the patients' relatives, but as far as it is known, they never got tested.
- Therapeutic strategy? Did it change (and how?) after the results of the genetic testing?
(Please, see lines 188-191 in a file “jcm-2537243 - editing can be seen”) However, the therapeutic strategy did not change after the diagnosis and resection of the affected organ continues to be the best suitable treatment method. In addition, the patient’s first presentation in the small bowel and relative sparing of the colon allowed for long-term avoidance of short bowel syndrome.
- Germline testing? Please describe the methodology and other findings of NGS testing. Which databases did you check to confirm that this finding is novel? What are the requirements that should be met to describe this finding as pathogenic with only one patient reported so far?
(Please, see lines 169-188 in a file “jcm-2537243 - editing can be seen”) The patient was referred for genetic counseling and molecular genetic testing was performed using Illumina Miseq next generation sequencing technology with targeted sequencing of colorectal cancer related genes (MLH1, MSH2, MSH6, PMS2, EPCAM, APC, MUTYH, STK11, SMAD4, BMPR1A TP53, NTHL1, POLE, POLD1, MSH3) included in TruSight Cancer panel, complemented by bioinformatic analysis. A deleterious pathogenic gene MSH2 (NM_000251.2) variant (mutation) c.1774_1775insT was detected. This alteration, located in coding exon 12 of the MSH2 gene, results from an insertion of one nucleotide at position 1774, causing a translational frameshift with a predicted alternate stop codon (p.Met592Ilefs*6). This variation of the coding sequence is expected to result in loss of function by premature protein truncation or nonsense-mediated mRNA decay and allele frequency not found in normal population control GnomAD database. As such, this alteration is interpreted as a disease-causing mutation and classified as pathogenic, Class 5 mutation according to ACMG criteria. This variant is novel, since was not previously reported in ClinVar, InSight databases or literature. Additionally, a heterozygous monoallelic MUTYH (NM_001128425.2) variant c.1187G>A (p.Gly396Asp) was detected, which is associated with MUTYH-associated polyposis only in bi-allelic (homozygous/compound heterozygous) [17]. This data confirmed Lynch syndrome diagnosis on a molecular level, clinically manifesting as type II.
- Have you reported this finding to databases like Clinvar?
Thank you for the question. We will add the mutation to the ClinVar database.
- The pathology images in the manuscript are very informative. Please discuss how the pathological characteristics of these tumors match (or not) the characteristics/phenotype that are suggestive of MMR-deficient tumors?
(Please, see lines 312-342 in a file “jcm-2537243 - editing can be seen”) The histological features suggestive of microsatellite instability in a colorectal adenocarcinoma are tumor-infiltrating lymphocytes (TILs), so-called „Crohn‘s like“ peritumoral transmural lymphocytic reaction, extracellular mucin pools, medullary growth pattern or signet-ring cells (or otherwise poorly differentiated/undifferentiated areas), and lack of tumor budding. The patient reported herein had a total of five colorectal tumors. The histology of the first one, a rectal carcinoma of 2001, was not reviewed since the slides were not available anymore. Yet the description in the pathology report mentions both small areas of solid growth and mucin production, at least hinting at the possibility of MMRP-deficient tumor. The second one, a splenic flexure carcinoma (Fig. 1, panel 1a) was a nodular tumor composed of solid neoplastic cell sheets, or areas of discohesive cells with rhabdoid features and tumor-infiltrating lymphocytes – a classic MSI tumor pattern. The three simultaneous tumors of 2014 had mixed features: the G3 tumor with a single lymph node metastasis had both abundant extracellular mucin production and signet ring cells (Fig. 1, panel 1b), while the other two were more conventional gland-forming adenocarcinomas, yet extracellular mucins were also present in lesser quantities (Fig. 1, panel 1c).
The histologic features that are specifically suggestive of MSI sebaceoma have not been extensively described, yet the very presence of a sebaceous neoplasm of the skin raises a possibility of Muir-Torre syndrome, a variant of Lynch syndrome. Yet this option was dismissed due to the overwhelming gastrointestinal presentation of tumors.
Urothelial MSI tumors share the presence of TILs with MSI colorectal carcinoma, but there are also distinct characteristics that can be suggestive of microsatellite instability. MSI urothelial carcinomas can have an inverted growth pattern, a higher grade, conspicuous nucleoli, brisk mitotic activity; some of the tumors might have heterogenous areas. The urothelial carcinoma described here was a G3 tumor with focal myxoid stroma, an admixture of signet ring cells, and focal perineoplastic lymphocytic infiltration, thus exhibiting at least some features suggestive of MSI.
Lastly, the list of features suggestive of MSI endometrioid carcinoma is mostly composed of the ones described above and includes TILs, solid areas, mucinous differentiation, higher grade, and tumor heterogeneity. Nevertheless, MSI endometrioid carcinomas often present as conventional G1 tumors as was in this case.
- Please include MLH1/MSH6 immunohistochemistry staining.
MSH6 immunohistochemistry is included in Fig. 1, panel 2c.
MLH1 staining is not available, as the original archived slide of MLH1 is of poor quality and has faded – as the tissue might give a false impression of lost expression, we have decided not to include it in the Figure 1. After the diagnosis of Lynch syndrome was established, further immunohistochemical staining was often limited to MSH2/MSH6 to confirm the known MMRP loss pattern.
- Does the development of sebaceoma suggest Muir-Torre syndrome? Please discuss.
(Please, see lines 222-228 in a file “jcm-2537243 - editing can be seen”) In 2020, sebaceoma of the scalp was detected and confirmed by the pathologists. Immunohistochemical analysis revealed the absence of MSH2 and MSH6 expression in the sebaceoma. Development of sebaceoma together with colorectal cancer and MSH2 gene mutation correctly suggest Muir-Torre syndrome. However, cutaneous presentations in this patient are minimal in comparison to internal organ malignancies. Muir-Torre syndrome was not diagnosed because of overwhelming gastrointestinal presentation of tumors.
- 2014 ascending colon resection with two cancers (one staged as N0, the other as N1). How is this possible given that the regional lymph nodes are the same for each tumor?
(Please, see lines 162-164 in a file “jcm-2537243 - editing can be seen”) In 2014, the resected colon had not two, but three tumors. Two of them were more or less conventional G2 gland-forming adenocarcinomas, while the third one was a G3 tumor with signet-ring cells. Since only one lymph node had a metastasis of purely signet-ring cells, the G3 tumor was interpreted as N1a, while the other two tumors were considered N0 to avoid overstaging.
- line 241 MLH6 mutation?
(Please, see lines 303-304 in a file “jcm-2537243 - editing can be seen”) The incorrect sentence was removed.
Reviewer 2 Report
Congratulations on the present paper!
Lynch syndromee represents an intensely debated subject.
There are few aspects that should be improved.
1. The introduction section is too long and also presents some repetitions.
2. The MSH2 mutation, which is the main subject of the manuscript is not sufficiently debated.
3. Sentences starting with words such as "We present", and "Our patient" should be changed for a more professional aspect.
4. The conclusion section should be more concise.
Minor English issues.
Author Response
- The introduction section is too long and also presents some repetitions.
Changes have been made. (Please, see line 36-98 in a file “jcm-2537243 - editing can be seen”) Some sentences were removed from the lines 65-67, 69-71, 90-93.
- The MSH2 mutation, which is the main subject of the manuscript is not sufficiently debated.
(Please, see lines 43-62 in a file “jcm-2537243 - editing can be seen”) The primary genes associated with LS are MLH1, MSH2, MSH6, and PMS2. While MLH3 and PMS1 have also been implicated in the disorder, their precise roles remain less well-defined [3]. All these six genes, MLH1, MSH2, MSH6, PMS2, MSH3, and MLH3, play a role in mismatch repair (MMR) [4]. Mismatch repair (MMR) system is important for maintaining genome integrity. MMR system can detect misincorporated nucleotides and fix these mistakes by excision or resynthesis process [5]. Microsatellite instability can be corrected by the MMR system as well [5]. However, when a mutation in genes encoding MMR system proteins occurs, microsatellite instability can be detected and development of various cancers can be implicated. Microsatellite instability (MSI) is found in 10-15% of all colorectal cancer (CRC) cases [6]. 3% of these cases are associated with Lynch syndrome [7].
MLH1 can be associated with both germline and somatic mutations, while MSH2, MSH6, and PMS2 are only related to genetic inheritance [8]. In order to detect changes in MMR protein different techniques can be used such as: immunohistochemistry (IHC), polymerase chain reaction (PCR), and large panel next-generation sequencing [8]. MMR proteins function as dimers. MLH1 forms a complex with PMS2 and MSH2 forms a complex with MSH6. Consequently, mutations in MLH1 result in immunohistochemical loss of staining in both MLH1 and PMS2. The same situation can be seen in mutation occurring in MSH2 that will result in loss of staining in MSH2 and MSH6. However, mutations in PMS2 and MSH6 only affect the mentioned gene’s product [8].
- Sentences starting with words such as "We present", and "Our patient" should be changed for a more professional aspect.
Changes have been made, please see lines 31, 250, 257, 266, 277, 282, 287, 348, 362, 363 in a file “jcm-2537243 - editing can be seen”.
- The conclusion section should be more concise.
Some sentences were removed. Please see lines 383-394 in a file “jcm-2537243 - editing can be seen”.
Round 2
Reviewer 1 Report
This is a much improved version of the manuscript. The authors addressed my concerns in a satisfactory way.
Specific point
Lines 76-77 This is not true. MLH1 promoter hypermethylation is the most common but somation mutations of the other genes have been also described. For example, see https://doi.org/10.1002/ijc.30820
Author Response
Lines 76-77 This is not true. MLH1 promoter hypermethylation is the most common but somation mutations of the other genes have been also described. For example, see https://doi.org/10.1002/ijc.30820
Thank you for Your comment.
We changed the sentence to "Both germline and somatic mutations are found in MMR genes, where MLH1 promoter hypermethylation is the most common somatic event (epimutation) [9], [10]." You can see it in lines 54-55.